# LANGUAGE-INTERFACED TABULAR OVERSAMPLING VIA PROGRESSIVE IMPUTATION AND SELF AUTHENTICATION

**June Yong Yang**[1,*] **Geondo Park**[1,*] **Joowon Kim**[1], **Hyeongwon Jang**[2], **Eunho Yang**[1,3]
KAIST[1], Seoul National University[2], AITRICS[3]
{laoconeth, geondopark, kjwispro, eunhoy}@kaist.ac.kr
janghw0911@gmail.com

## ABSTRACT

Tabular data in the wild are frequently afflicted with class-imbalance, biasing machine learning model predictions towards major classes. A data-centric solution to this problem is oversampling - where the classes are balanced by adding synthetic minority samples via generative methods. However, although tabular generative models are capable of generating synthetic samples under a balanced distribution, their integrity suffers when the number of minority samples is low. To this end, pre-trained generative language models with rich prior knowledge are a fitting candidate for the task at hand. Nevertheless, an oversampling strategy tailored for tabular data that utilizes the extensive capabilities of such language models is yet to emerge. In this paper, we propose a novel oversampling framework for tabular data to channel the abilities of generative language models. By leveraging its conditional sampling capabilities, we synthesize minority samples by progressively masking the important features of the majority class samples and imputing them towards the minority distribution. To reduce the inclusion of imperfectly converted samples, we utilize the power of the language model itself to self-authenticate the labels of the samples generated by itself, sifting out ill-converted samples. Extensive experiments on a variety of datasets and imbalance ratios reveal that the proposed method successfully generates reliable minority samples to boost the performance of machine learning classifiers, even under heavy imbalance ratios.

## 1 INTRODUCTION

Tabular data is one of the most common forms of data in real-world applications, spanning vast industries such as healthcare (Johnson et al., 2021), marketing (Sakar et al., 2019), and finance (Shah et al., 2022). However, due to their idiosyncratic nature and volatility in the data collection process, tabular data in the wild are often ridden with class-imbalance. For example, a financial report dataset curated to predict a major economic collapse is likely to be class-imbalanced as such events are rare in the real world. In turn, machine learning classifiers trained from such imbalanced data are inclined to be biased, as the model is trained towards classifying an overly significant amount of samples as the majority class. This misalignment in generalization directly impacts the classification performance of minority classes, which are often the critical foci of interest. For instance, a cancer prediction classifier would be of limited use if its predictions are substantially inclined towards predicting all patients as cancer-free, as resulting false negatives inevitably incur major liabilities.

Faced with the issue, diverse methodologies have been proposed to address the issue of class imbalance. These methods predominantly focus on either modifying the model itself (Kang et al., 2020; Tian et al., 2020; Menon et al., 2021; Hong et al., 2021) or adapting its loss function to enhance robustness against class imbalance (Cao et al., 2019). However, these approaches fall short in terms of general applicability since they require the end-user to directly modify the existing off-the-shelf machine learning models. Considering the common practice of an end-user to employ a range of readily available models depending on the domain peculiarities of the given data, these model-

---

*Equal contribution.

centric strategies are bound to incur potential difficulties and friction from the perspective of the end-user. Consequently, a model-agnostic, data-centric approach of *minority oversampling* (Chawla et al., 2002; Han et al., 2005; He et al., 2008) has been proposed to synthetically generate and add minority samples to balance the class distribution. Through such means, the model itself or its learning mechanism can be kept intact while the data itself becomes balanced to remedy the issue.

Recent advances in deep generative models have bestowed the means to generate high-quality synthetic data (Kingma & Welling, 2013; Goodfellow et al., 2020; Ho et al., 2020; Radford et al., 2019). Given their success in other data modalities (Rombach et al., 2022; OpenAI, 2023), they have been successfully adapted to the tabular domain (Choi et al., 2017; Xu et al., 2019; Kim et al., 2023; Kotelnikov et al., 2023; Borisov et al., 2023). By training these generative models on the imbalanced dataset, synthetic minority samples can be drawn from the generative distribution to oversample the minority classes. However, this approach often struggles in practical situations when the number of minority samples is too small to successfully train a generative model. In this situation, the model may fail to learn the minority distribution, or simply memorize the minority samples.

To this end, a stronger generative model equipped with pre-trained knowledge is required to mitigate overfitting. An intriguing solution to this problem is the usage of generative large language models (LLMs) (Radford et al., 2019; OpenAI, 2023; Touvron et al., 2023a;b) to process tabular data through language interfaces (Dinh et al., 2022). Recent studies have demonstrated that by representing tabular data as *free text* and leveraging the knowledge of language models, effective task performance can be achieved even when the amount of data is limited (Hegselmann et al., 2023; Nam et al., 2023). However, this abrupt shift of paradigm makes it challenging to apply conventional imbalance handling techniques, such as borderline sampling (Han et al., 2005; He et al., 2008), major-to-minor translation (Kim et al., 2020), or style transfer (Kim et al., 2022). Thus, there emerges a need for a systematic framework to effectively perform minority oversampling based on language models.

In this paper, we propose Language-Interfaced Tabular Oversampling (LITO), a novel oversampling framework for tabular data that comprehensively utilizes the power of language-interfaced tabular learning. Using the conditional sampling capabilities of the generative language model, samples with minority traits can be synthesized through class label prompting. Based on this functionality, we develop a borderline sampling method that converts a majority sample to a minority by progressively 'puncturing' the feature values of the sample and imputing them under minority conditioning, so that they convey the characteristics of the minority class.

However, recent observations on the generative capabilities of language models report the potential for biased (Gallegos et al., 2023) or contradictory (Ji et al., 2023) generation. When supplied with class-imbalanced data, the model might incline towards generating majority classes even with class conditioning, introducing noisy samples. Such risks can be exacerbated during the execution of aggressive techniques such as borderline sampling, as there is a possibility that the sample may not be fully converted. To effectively sample synthetic minorities under these conditions, we propose a simple yet effective rejection sampling procedure to prevent the inclusion of ill-synthesized minority samples. Motivated by self-alignment methodologies (Sun et al., 2023) of large language models, we use the language model on itself to predict the labels of its own generated samples, filtering out ill-synthesized suspects. Integrating these components, we then propose a progressive imputation scheme harnessing the power of the language model itself to guide its own oversampling procedure. We validate our method against various tabular datasets from OpenML-CC18 (Bischl et al., 2019) and UCI machine learning repository (Dua & Graff, 2017), with varying imbalance ratios. Also, we demonstrate the applicability of the LITO framework on black-box chatbot LLMs such as GPT3.5-turbo through in-context learning (ICL).

Our contributions are threefold:

- We propose a novel tabular oversampling strategy based on generative language models that converts a majority sample to a minority sample by inducing missing values in the important columns and filling them through minority-class conditioned imputation.

- To mitigate the problem of faulty synthesis, we introduce a simple yet effective technique that utilizes the generative language model itself to filter out defective samples.

- Our oversampling strategy enhances the performance of off-the-shelf machine learning models, even for highly imbalanced datasets.

## 2 BACKGROUND

### 2.1 OVERSAMPLING FOR CLASS-IMBALANCED DATA

We formally define the problem setting of our interest. We take into consideration a supervised learning problem given a training dataset $D := \{z_n = (x_n, y_n)\}_{n=1}^N$, where $x \in \mathbb{R}^m$ is the feature vector and $y \in \{1, \ldots, C\}$ is the class label of the data point $z$. For each class, we denote the number of training samples that belong to class $c$ as $N_c$, where the classes are sorted by their number of samples in descending order without the loss of generality as in $N_1 \geq N_2 \geq \cdots \geq N_C$. A dataset is *class-imbalanced* if the number of samples for each class is not equal and skewed: $N_1 \gg N_C$. We define the *majority* class as the class with the largest number of samples ($N_1$) and other classes as *minority* classes. The *imbalance ratio* is defined as $\alpha = \frac{N_1}{N_C}$ An *oversampling* strategy introduces synthetic minority samples to the dataset until the number of samples for all minority classes becomes equal to the majority class, as in $N_1 = \tilde{N}_2 = \cdots = \tilde{N}_C$. This process yields a class-balanced dataset $\tilde{D}$ which in turn is employed to train a desired machine learning model.

### 2.2 LANGUAGE-INTERFACED TABULAR GENERATION

To the best of our knowledge, a clear consensus on the precise definition of tabular data is yet to be reached. In this paper, we temporarily define tabular data as data formatted into $N$ rows (samples) and $M$ columns (features), which can be cast into the format of comma-separated values (CSV). Unlike other data modalities such as vision, speech, or graphs, a notable characteristic of tabular data is that it can be straightforwardly represented in free text. This suggests the possibility of applying natural language processing algorithms to process tabular data. Particularly, an intriguing approach is the handling of tabular data using generative language models. Since tabular data can be readily formatted into text, they can be processed by generative language models without the usage of external adapters or representation alignment (Tsimpoukelli et al., 2021). Toward this goal, Dinh et al. (2022); Hegselmann et al. (2023); Borisov et al. (2023) explored the prediction and generation capabilities of generative language models by transforming tabular data into text prompts using textual encoders. Following Dinh et al. (2022), we refer to such a paradigm as *language interfacing* for tabular data, and refer to the core language model as *tabular language models* (TLM), for the scope of this paper. Given a tabular dataset, the $n$-th row of the table can be represented as a textual prompt $\mathbf{t}_n$ using the following rule:

$$t_{n,m} = [h_m, \text{``is''}, v_{n,m}, \text{``,''}] \text{ and } \mathbf{t}_n = [t_{n,1}, t_{n,2}, \ldots, t_{n,M}] \tag{1}$$

where $n \in \{1, \ldots, N\}$, $m \in \{1, \ldots, M\}$, $v_{n,m}$ is the $(n, m)$-th value of the table, and $h_m$ is the name of the $m$th column. The row index $n$ may be omitted for brevity. Here, we follow the encoding format proposed in Borisov et al. (2023). However, the syntactic structure or the choice of articles to construct the textual prompt may vary.

In the same vein, the generation of tabular data is also feasible using the language-interfacing paradigm. The GReaT (Borisov et al., 2023) framework for tabular generation fine-tunes an autoregressive pre-trained transformer with tabular data to generate novel tabular samples. A textual encoder encodes the column names and features into text, which is then fed to the language model for fine-tuning. Since the columns of the table do not exhibit spacial locality, a random feature order permutation function is employed. To generate synthetic samples, conventional autoregressive sampling is used. Since the generative model is fine-tuned with random feature order permutation, arbitrary conditional sampling is possible by prompting the model with conditioning column names and values to a TLM $\mathbf{f}$:

$$\tilde{\mathbf{t}} := \left[\tilde{t}_{s_1}, \tilde{t}_{s_2}, \ldots, \tilde{t}_{s_{M-l}} \mid t_{r_1}, t_{r_2}, \ldots, t_{r_l}\right] = \mathbf{f}([t_{r_1}, t_{r_2}, \ldots, t_{r_l}]) \tag{2}$$

where $t_r$ are column conditional prompts and $t_s$ are sampled column texts. Such capabilities can be readily utilized for class-conditioned sampling by placing the label column and values in the front of the prompt for autoregressive decoding. Although certain deep generative models for tabular data support class conditional sampling, arbitrary feature-conditioned sampling capabilities are generally not present in other forms of tabular generative models.

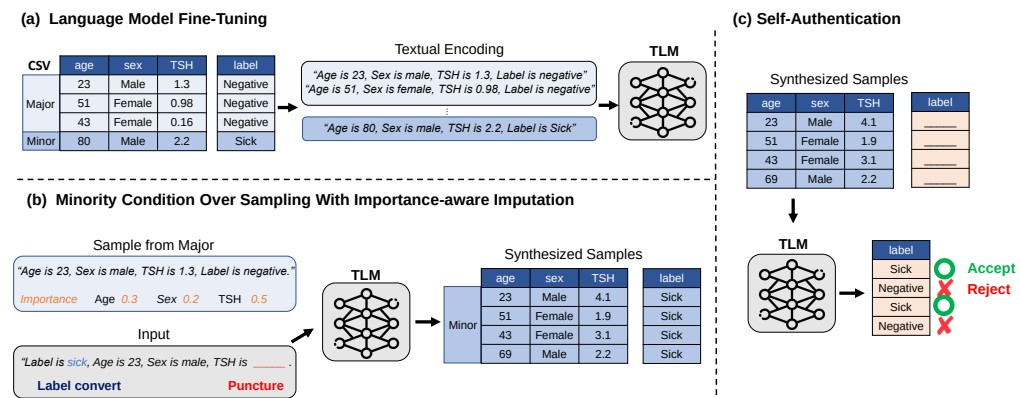

Figure 1: Overview of our framework (LITO). Using a TLM backbone, minority candidates are generated via class-conditioned borderline sampling. Then, the generated candidates are self-authenticated by re-imputing their labels. For rejected candidates, the process is repeated until the sample is converted as long as the prediction confidence increases with each iteration.

## 3 LANGUAGE-INTERFACED TABULAR OVERSAMPLING (LITO)

In this section, we introduce our Language-Interfaced Tabular Oversampling (LITO) framework that synthesizes minority samples by leveraging the conditional sampling capabilities and the rich prior knowledge of pre-trained generative language models. Our framework is composed of two components: minority class-conditioned sampling to generate synthetic minority candidates (Section 3.1), and the self-authentication procedure to filter out defective samples (Section 3.2). The overall framework of LITO is described in Figure 1.

### 3.1 MINORITY-CONDITIONED SAMPLING WITH IMPORTANCE-AWARE IMPUTATION

The first stage of our framework is to generate minority-class conditioned samples via class-conditioned prompting, using arbitrary conditioned sampling (Equation 2) of TLMs. A synthetic minority sample of class $c$ can be generated from a TLM $\mathbf{f}$ via class-conditioning by prompting the model with the minority class label:

$$\mathbf{t}^c = \mathbf{f}([t^c_{label}]) \tag{3}$$

where $t^c_{label} = [\text{"}label\text{"}, \text{"}is\text{"}, \text{"}c\text{"}, \text{"},\text{"}]$. With this functionality, we propose a novel borderline sampling strategy tailored to a decoder-only TLM by converting a majority sample to a minority sample through class-conditioned imputation. First, a substrate majority class sample $s$ is randomly drawn from the entire training set. Subsequently, we induce missing values by 'puncturing' out $k$ columns, obtaining a truncated prompt $[t_1, t_2, \ldots, t_{M-k}]$. By prompting the sample as the target minority class and performing conditional imputation, we convert the sample to the targeted minority class:

$$\mathbf{t}^c = \mathbf{f}([t^c_{label}, t_1, t_2, \ldots, t_{M-k}]) \tag{4}$$

where the conditioned columns are re-indexed without the loss of generality. Note that our approach is equivalent to full class-conditioned sampling when $k = M$.

Considering the heterogeneity of columns, randomly puncturing and imputing the feature values may neglect important columns that are key towards defining a sample as a majority class or a minority class. In other words, imputing unimportant columns may not be sufficient to convert a majority class sample to a minority class sample. With this in mind, we propose to puncture and impute columns guided by a feature importance criterion. In detail, we utilize the self-attention scores of the TLM to attribute the importance of column features. We input a given substrate $s$ to the TLM and extract the last-layer attention map to obtain the attention scores with respect to the label token, and calculate the importance score for each tabular column by summing the attention scores of the tokens that consist of the respective column. In the case where the label consists of multiple tokens, we use the mean of the attention scores on the label tokens. The importance score of the

$m$-th column of substrate sample $s$ is given as:

$$I(s,m) = \frac{1}{|\text{tok}(t_{s,label})|} \sum_i^{\text{tok}(t_{s,label})} \sum_j^{\text{tok}(t_{s,m})} \sum_h A_{i,j,h}^s \qquad (5)$$

where $\text{tok}(t_{s,m})$ is the set of tokenized sequence indices that consists of the $m$-th tabular column of substrate $s$, $h$ is the attention head index, and $A_{i,j,h}^s$ is the last layer attention score of the $i$-th token by the $j$-th token for head $h$ for substrate $s$. After obtaining the sample-wise column importance scores, we select top-$k$ important columns to be subject to imputation.

### 3.2    REJECTION SAMPLING VIA SELF-AUTHENTICATION

Although intuitive, the proposed strategy carries inherent risks due to the uncertainty of whether the samples generated by class-conditional prompting or borderline sampling are indeed minority samples. For class-conditioned sampling, there is a possibility that the model may generate a majority class sample even under such conditioning if the model is inclined towards the majority class. For borderline sampling, if the number of column punctures is insufficient, there remains a possibility for the sample to either retain its majority class identity or fall into an ambiguous category between the two. Despite the provision of class-conditional prompting, oversampling through imputation can still yield either ambiguous minority samples or even majority samples, which can adversely affect the formation of decision boundaries. Therefore, it is imperative to devise a method to filter out such ill-generated synthetic samples.

For this purpose, we propose a simple yet effective rejection sampling method that utilizes the characteristics of language models. Our key intuition is that the generative language model is also capable of imputing the *label* of the given sample when the label is absent. Thus, a generated sample can be verified by the language model itself by imputing the label of the sample:

$$\left[ \tilde{t}_{label} \mid t_1, \ldots, t_M \right] = \mathbf{f}([t_1, \ldots, t_M]) \qquad (6)$$

Based on this functionality, we introduce the Self-Authentication procedure to the oversampling process. After a minority sample is generated, we discard the label of the generated sample and re-impute the label using the TLM. If the label of the synthesized minor is indeed predicted as a minor, then the sample is accepted. If else, then the sample is deemed faulty and is discarded. This process is akin to the self-alignment process of language models (Sun et al., 2023), where the language model itself is utilized to control the generation process of the language model. We refer to this sample-and-authenticate framework as LITO. Note that our self-authentication method not only can be combined with borderline sampling, but also full class-conditional sampling (LITO-C).

### 3.3    ADAPTIVE OVERSAMPLING WITH PROGRESSIVE IMPUTATION

Combining the components described above, we now propose LITO-B, an adaptive borderline sampling strategy. When transforming the features of majority class samples into minority class samples through imputation, it is important to acknowledge that the number of column imputations required to successfully convert a majority class sample into the target minority class may vary from one sample to another. In other words, certain samples might not achieve proper translation due to an insufficiently determined number of column punctures. In such cases, despite the potential for these samples to be converted into the minority class by increasing the number of column punctures, they are preemptively excluded during self-authentication.

To synthesize more diverse and probable samples, we propose a progressive imputation sampling method, wherein we incrementally increase the column puncture count while iteratively generating and filtering samples. For each round, we puncture $k$ columns according to the importance ordering and execute conditional imputation. Then, self-authentication is performed to confirm the converted samples. For the samples that have failed the authentication, we puncture the next $k$-important columns and iterate the process. Additionally, if the prediction probability of the target label (confidence) does not increase after each imputation round, then the substrate sample is discarded. Through this process, the number of imputed columns can be adapted dynamically, generating diverse samples. The overall process is described in Algorithm 1.

---

**Algorithm 1** LITO-B

---

1: **Input:** Training set $D := \{z_n = (x_n, y_n)\}_{n=1}^N$, Fine-tuned TLM $f$, Number of classes $C$, puncture count per round $k$, Authenticated_samples $\leftarrow$ []
2: **for** each class $c_i$ in $C$ **do**
3:     **while** generate amount $== |c_{\mathrm{major}}| - |c_i|$ **do**
4:         Sample the substrate $s$ in $D$ where $y_s \neq c_i$
5:         Compute the attention score $I$ of each columns from equation 5
6:         Sort $I$, by descending order
7:         $o = $ Index of $I$
8:         round $r = 0$
9:         **while** True **do**
10:             Puncture $o[r : (r + 1) \times k]$ indices at $s$ and change label column to target label $c_i$
11:             Resulting in $s_p = [t_{label}^{c_i}, t_1, t_2, \ldots, t_{M-k}]$                     ▷ Imputation
12:             Synthesized sample $s_r = f(s_p)$
13:             Puncture *label* column of $s_r$                     ▷ For self-authentication
14:             $pred, logit = f(s_r)$
15:             **if** $pred == c_i$ **then**
16:                 Authenticated_samples $\leftarrow s_r$
17:                 Break
18:             **else**
19:                 **if** $prob(c_i, s_r) >= prob(c_i, s_{r-1})$ **then**       ▷ Check confidence increase
20:                     $s \leftarrow s_r$
21:                     $r+ = 1$
22:                 **else**
23:                     Break
24:                 **end if**
25:             **end if**
26:         **end while**
27:     **end while**
28: **end for**

---

### 3.4 EXTENDING TO BLACK-BOX LANGUAGE MODELS THROUGH IN-CONTEXT LEARNING

Considering the scaling law of LLM performance with respect to their number of parameters (Kaplan et al., 2020), it can be projected that the effectiveness of the LITO framework will also increase with the size of the backbone TLM. However, for powerful but proprietary black-box LLMs, fine-tuning a model to perform tabular oversampling entails practical difficulties for an end-user. Intriguingly, recent variants of LLMs possess the ability to effectively learn from the input data in the inference phase through *in-context learning*, enabling an end-user to adapt LLMs to their distinctive needs without fine-tuning the language model. By supplying an LLM with the imbalanced dataset and instruction prompts for feature importance calculation, conditional imputation, and self-authentication (Appendix C), the LITO framework can be adapted to black-box LLMs, as we demonstrate in Section 4.5.

## 4 EXPERIMENTS

In this section, we demonstrate the effectiveness of our framework by conducting experiments on multiple publicly available datasets. We validate our method and its individual components following the conventions of prior works. Below, we provide a brief overview of our experimental setting in Section 4.1, followed by empirical results and detailed analyses in the subsequent sections. We also analyze the sampling quality and diversity in Appendix B.

### 4.1 EXPERIMENTAL SETUP

**Datasets.** We validate our method on six tabular benchmark datasets: `Default`, `Shoppers`, `Sick`, and `Diabetes` for binary classification, `Obesity` and `Satimage` for multi-class classification. They exhibit variation in terms of sample sizes and feature dimensions. The detailed characteristics of the datasets are provided in Appendix A. We partition the datasets into 80% for training and 20% for the test set following the previous works (Kim et al., 2022). For datasets with rela-

Table 1: Comparison of our method with other baselines for binary classification tasks. We conduct experiments on both a mild class-imbalance setting ($\alpha = 10$) and an extremely class-imbalanced setting ($\alpha = 100$). We report the F1-score and balanced accuracy with the standard errors on four benchmark datasets. The best performance is **bolded**, and the second-best is underlined.

| | *Extreme imbalance (imbalance ratio $\alpha = 100$)* | | | | | | | |
|---|---|---|---|---|---|---|---|---|
| **Dataset** | Default | | Shoppers | | Sick | | Diabetes ($\alpha = 20$) | |
| | F1 | bAcc | F1 | bAcc | F1 | bAcc | F1 | bAcc |
| Vanilla | 45.16 ±0.09 | 50.60 ±0.04 | 54.02 ±0.11 | 54.15 ±0.06 | 67.71 ±0.79 | 66.13 ±0.48 | 49.54 ±0.23 | 54.29 ±0.09 |
| SMOTE | 55.25 ±0.13 | 57.41 ±0.08 | 64.72 ±0.41 | 63.40 ±0.06 | 68.90 ±0.30 | 69.65 ±0.45 | 61.06 ±0.54 | 61.28 ±0.42 |
| B-SMOTE | 51.64 ±0.13 | 53.89 ±0.07 | 64.65 ±0.20 | 62.15 ±0.22 | 48.68 ±1.45 | 58.18 ±1.58 | 59.77 ±0.47 | 60.46 ±0.38 |
| CTGAN | 55.13 ±0.16 | 55.47 ±0.13 | 57.25 ±0.14 | 56.07 ±0.09 | 59.26 ±0.53 | 59.00 ±0.85 | 55.31 ±1.40 | 55.50 ±1.32 |
| SOS | 56.19 ±0.31 | 56.71 ±0.19 | 65.65 ±0.52 | 63.30 ±0.45 | 80.53 ±0.15 | 83.04 ±0.35 | 49.80 ±0.32 | 54.39 ±0.13 |
| GreaT | 51.38 ±0.09 | 52.71 ±0.09 | 56.64 ±0.10 | 65.02 ±0.10 | 70.47 ±0.62 | 82.73 ±0.60 | 49.44 ±0.68 | 49.59 ±0.54 |
| LITO-C | 61.91 ±0.07 | 62.11 ±0.08 | 73.75 ±0.13 | 71.06 ±0.06 | 85.09 ±0.81 | 84.69 ±1.06 | 63.04 ±0.46 | 63.20 ±0.41 |
| LITO-B | **65.52** ±0.14 | **67.33** ±0.13 | **74.58** ±0.20 | **73.09** ±0.23 | **85.63** ±0.95 | **87.67** ±0.60 | **64.23** ±0.42 | **64.05** ±0.38 |
| | *Mild imbalance (imbalance ratio $\alpha = 10$)* | | | | | | | |
| **Dataset** | Default | | Shoppers | | Sick ($\alpha = 12.2$) | | Diabetes | |
| | F1 | bAcc | F1 | bAcc | F1 | bAcc | F1 | bAcc |
| Vanilla | 55.27 ±0.07 | 55.97 ±0.05 | 71.08 ±0.10 | 68.23 ±0.17 | 85.76 ±0.77 | 83.01 ±0.89 | 52.66 ±0.50 | 55.77 ±0.27 |
| SMOTE | 62.81 ±0.20 | 62.54 ±0.19 | 74.31 ±0.16 | 74.89 ±0.28 | 71.91 ±0.58 | 72.59 ±0.79 | 65.85 ±0.51 | 65.70 ±0.27 |
| B-SMOTE | 61.74 ±0.24 | 61.37 ±0.23 | 69.84 ±0.16 | 76.53 ±0.46 | 67.63 ±0.79 | 69.52 ±2.77 | 65.48 ±0.70 | 65.20 ±0.49 |
| CTGAN | 59.15 ±0.22 | 59.34 ±0.23 | 72.30 ±0.18 | 70.28 ±0.24 | 77.15 ±0.62 | 83.77 ±0.89 | 62.66 ±0.96 | 63.43 ±0.93 |
| TVAE | 63.23 ±0.15 | 61.70 ±0.12 | 71.24 ±0.22 | 69.00 ±0.26 | 82.02 ±0.47 | 86.38 ±0.57 | 63.17 ±1.04 | 62.77 ±0.91 |
| SOS | 60.96 ±0.21 | 60.16 ±0.20 | 74.24 ±0.29 | 75.19 ±0.30 | 84.04 ±0.40 | 90.51 ±0.19 | 53.06 ±0.42 | 55.92 ±0.23 |
| GreaT | 65.22 ±0.10 | 66.78 ±0.10 | 68.32 ±0.20 | 75.89 ±0.11 | 82.86 ±0.34 | 91.09 ±0.22 | 64.94 ±0.61 | 65.31 ±0.72 |
| LITO-C | **67.94** ±0.03 | 67.48 ±0.04 | **75.85** ±0.13 | 77.66 ±0.12 | **86.00** ±0.19 | 91.13 ±0.14 | 64.29 ±0.38 | 64.00 ±0.36 |
| LITO-B | 66.86 ±0.05 | **68.85** ±0.04 | 75.59 ±0.05 | 76.86 ±0.09 | 85.83 ±0.12 | **91.92** ±0.14 | **65.94** ±0.44 | **66.12** ±0.39 |

tively small sizes (`Diabetes`, `Sick`), we split the dataset into 70% training set and 30% test set. To construct class-imbalanced datasets, we deliberately reduce the number of training samples per class to establish a long-tail distribution in the class distribution. Then, given the parameter $\gamma$ which controls the imbalance ratio $\alpha = \gamma_l^{-\frac{l-1}{L-1}}$ of datasets, we decide the number of samples for class $l$ following $\tilde{N}_l = N_1 \cdot \alpha$ and the remainder are discarded. We conduct experiments on mild ($\alpha = 10$) and extremely class-imbalanced ($\alpha = 100$) settings.

**Baselines.** We select two statistical methods and four deep learning-based generative models as baselines. SMOTE (Chawla et al., 2002) and Borderline-SMOTE (B-SMOTE) (Han et al., 2005) are classical oversampling methods that augment the samples of minor classes. For deep generative models, we use CTGAN (Xu et al., 2019), TVAE (Xu et al., 2019), and SOS (Kim et al., 2022). CTGAN is a generative adversarial network (GAN) specifically designed to generate synthetic tabular data samples under conditional constraints. Similarly, TVAE is a variational autoencoder (VAE) modified for tabular data. We do not compare with TVAE in the case of extremely imbalanced setting ($\alpha = 100$), as the model exhibits mode-collapse behavior, completely failing to generate any minority samples. SOS is a score-based generative model equipped with imbalance-tailored techniques, proposed as an oversampling method for imbalanced tabular data. GReaT (Borisov et al., 2023) is a generative TLM based on fine-tuning GPT-2 (Radford et al., 2019) and its distilled version (Sanh et al., 2019). For the experimental evaluations below, we use GReaT based on Distill-GPT2 as the baseline. For our experimental evaluations, we adopt the same Distill-GPT2 GReaT model as our backbone TLM. Finally, Vanilla indicates the case where no oversampling is employed and the machine learning classifiers are trained with the imbalanced train set as-is.

**Evaluation protocol.** Our primary evaluation metric to measure the strength of a given oversampling strategy is *machine learning (ML) efficiency*. In detail, ML efficiency quantifies the classification performance of a given set of machine learning models trained on a training set augmented with oversampled data, which includes the original training samples and the synthesized samples evaluated on the held-out test set. Following the common evaluation procedure in the literature (Kim et al., 2022; Borisov et al., 2023), we compute the average classification performance of a curated set of ML models. For binary classification, we employ Decision Trees (Loh, 2011), AdaBoost (Schapire et al., 1999), Logistic Regression (Cox, 1958), and Multi-Layer Perceptron (Bishop & Nasrabadi, 2006), following (Kim et al., 2022). For multi-class classification, we add Random Forest (Breiman, 2001) and XGBoost (Chen & Guestrin, 2016) and exclude AdaBoost and MLP due to their observed training instabilities. For evaluation, minority classes are generated and added to the training

Table 2: Comparison of our method with other baselines for multi-class classification tasks. We conduct experiments on both mild class imbalance settings ($\alpha = 10$) and extremely class-imbalanced settings ($\alpha = 100$). We report the F1-score and balanced accuracy with standard errors on two benchmark datasets. The best performance is **bolded**, and the second-best is underlined.

| Dataset | Extreme imbalance ($\alpha = 100$) | | | | Mild imbalance ($\alpha = 10$) | | | |
| | Obesity | | Satimage | | Obesity | | Satimage | |
| | F1 | bAcc | F1 | bAcc | F1 | bAcc | F1 | bAcc |
|---|---|---|---|---|---|---|---|---|
| Vanilla | 48.69 $\pm 0.18$ | 52.06 $\pm 0.15$ | 70.43 $\pm 0.05$ | 71.70 $\pm 0.04$ | 71.39 $\pm 0.09$ | 71.32 $\pm 0.12$ | 80.52 $\pm 0.07$ | 80.04 $\pm 0.07$ |
| SMOTE | 34.59 $\pm 0.80$ | 39.90 $\pm 0.65$ | 77.63 $\pm 0.32$ | 77.34 $\pm 0.27$ | 49.61 $\pm 1.76$ | 54.21 $\pm 1.24$ | 84.03 $\pm 0.23$ | 83.36 $\pm 0.22$ |
| B-SMOTE | 37.39 $\pm 1.15$ | 41.79 $\pm 0.95$ | 74.53 $\pm 0.14$ | 74.15 $\pm 0.15$ | 50.58 $\pm 0.97$ | 54.22 $\pm 1.05$ | 82.21 $\pm 0.18$ | 81.69 $\pm 0.30$ |
| CTGAN | 62.67 $\pm 0.97$ | 63.42 $\pm 0.94$ | 65.24 $\pm 0.46$ | 68.05 $\pm 0.34$ | 62.43 $\pm 0.68$ | 62.91 $\pm 0.48$ | 78.07 $\pm 0.41$ | 77.64 $\pm 0.34$ |
| TVAE | - | - | - | - | 69.75 $\pm 0.94$ | 70.33 $\pm 0.95$ | 82.17 $\pm 0.30$ | 81.69 $\pm 0.30$ |
| SOS | 47.53 $\pm 0.74$ | 50.79 $\pm 0.52$ | 75.66 $\pm 0.28$ | 76.35 $\pm 0.24$ | 68.66 $\pm 0.89$ | 69.93 $\pm 0.86$ | 83.05 $\pm 0.35$ | 82.79 $\pm 0.32$ |
| GreaT | 57.18 $\pm 0.45$ | 58.20 $\pm 0.60$ | 74.32 $\pm 0.19$ | 74.64 $\pm 0.20$ | 76.73 $\pm 0.24$ | 77.98 $\pm 0.12$ | 82.36 $\pm 0.17$ | 82.43 $\pm 0.16$ |
| LITO-C | 66.62 $\pm 0.47$ | 67.58 $\pm 0.56$ | **81.74** $\pm 0.13$ | **81.14** $\pm 0.12$ | 78.13 $\pm 0.24$ | 79.23 $\pm 0.12$ | 84.53 $\pm 0.05$ | 83.82 $\pm 0.05$ |
| LITO-B | **68.02** $\pm 0.44$ | **69.08** $\pm 0.49$ | 76.81 $\pm 0.16$ | 78.24 $\pm 0.18$ | **79.74** $\pm 0.44$ | **80.64** $\pm 0.36$ | **84.69** $\pm 0.05$ | **83.92** $\pm 0.11$ |

set using the respective oversampling methods until all the minority classes reach the number of the majority class. To account for the randomness in the sampling procedure, we repeat the oversampling process 4 times, and train the machine learning models 5 times for a single oversampling instance, resulting in a total of 20 evaluations per method. We measure the F1-score and balanced accuracy (bAcc) and report their mean and standard error, as the uniform split of the test set is also imbalanced. A more detailed description of the experimental setting is described in Appendix A.

## 4.2 BINARY CLASSIFICATION

We first evaluate the effectiveness of our method in addressing imbalanced binary classification tasks. For experiments on mild imbalance scenarios, we use $\alpha = 10$ except for `Sick` since its natural imbalance ratio is $\alpha = 12.2$. For extreme imbalance experiments, we use $\alpha = 100$ except for `Diabetes` where we use $\alpha = 20$, as all generative models collapse for $\alpha = 100$. The results are presented in Table 1. For both mild imbalance and extreme imbalance scenarios, our methods consistently outperform all baselines, including statistical and deep generative methods, on all four tabular datasets. These results demonstrate the effectiveness of the minority samples synthesized through our oversampling strategy in assisting machine learning algorithms in their formation of more effective decision boundaries. As demonstrated by the significant improvement observed with our method in the case of highly imbalanced cases, our method is able to effectively perform even in extremely class-imbalanced scenarios with a limited number of samples for the minority class. Notably, comparing our method to GreaT, we observe a significant performance difference, implying the importance of self-authentication.

## 4.3 MULTI-CLASS CLASSIFICATION

To validate the effectiveness of our method on imbalanced multi-class classification tasks, we also conduct experiments on two multi-class tabular benchmark datasets. Note as there is more than one minority class in multi-class imbalanced data, a wide variety of one-to-one imbalance ratios exist within the dataset. As shown in Table 2, our method brings better imbalance handling performance in most cases compared to other baselines. In the extreme imbalance setting, our methods clearly outperform all baselines by large margins. For mild imbalance scenarios, our method consistently outperforms other baselines in both cases.

## 4.4 ABLATION STUDY

Here, we conduct an ablation study to verify the effect of the individual components that comprise the LITO framework: importance-aware imputation, self-authentication, and progressive imputation. For `Shoppers` and `Sick` datasets with imbalance ratio $\alpha = 100$, we compare the ML efficiency performance by incrementally adding the respective components, as shown in Table 3. First, we observe that importance-aware imputation increases the performance over random imputation. Second, self-authentication significantly boosts the performance. Finally, progressive imputation improves over single-iteration imputation. These results confirm the contributions of each LITO component.

Table 3: Ablation study investigating the individual components of LITO. We analyze the effects of adding importance-aware imputation, self-authentication, and progressive imputation.

| Sampling rounds | Puncture criterion | Self-authentication | Shoppers | | Sick | |
|---|---|---|---|---|---|---|
| | | | F1 | bAcc | F1 | bAcc |
| Single iteration | Random | ✗ | 53.65 ±0.33 | 63.32±0.38 | 65.70 ±1.03 | 77.68 ±1.03 |
| | Importance | ✗ | 54.31 ±0.19 | 64.64 ±0.26 | 67.13 ±0.79 | 79.64 ±0.38 |
| | Importance | ✓ | 73.39 ±0.16 | 73.08 ±0.16 | 84.53 ±0.70 | 84.93 ±0.60 |
| Progressive | Random | ✓ | 73.57 ±0.09 | 70.58 ±0.07 | 82.74 ±0.20 | 86.29 ±0.40 |
| | Importance | ✓ | **74.58** ±0.20 | **73.09** ±0.23 | **85.63** ±0.95 | **87.67** ±0.60 |

## 4.5 In-context LITO with Black-Box Language Models

Section 3.4 discussed the possibilities of adopting LITO to larger language models via in-context learning (ICL) and prompting. Here, we conduct a proof-of-concept experiment to demonstrate the performance of in-context LITO using OpenAI `GPT-3.5-turbo` API. For the highly imbalanced ($\alpha = 20$) setting of the `diabetes` dataset, we report the performance of LITO-C and LITO-B against the baselines. Table 4 shows that oversampling minority class samples through in-context learning is indeed effective. The detailed settings and additional experiments are provided in Appendix C.

Table 4: Performance of in-context LITO for the `diabetes` dataset ($\alpha = 20$).

| Method | F1 | bAcc |
|---|---|---|
| Vanilla | 49.54 | 54.29 |
| SMOTE | 61.06 | 61.28 |
| B-SMOTE | 59.77 | 60.46 |
| CTGAN | 55.31 | 55.50 |
| SOS | 49.80 | 54.39 |
| GReaT (Distill-GPT2) | 49.44 | 49.59 |
| ICL-LITO-C (GPT3.5) | **67.37** | **66.88** |
| ICL-LITO-B (GPT3.5) | 63.68 | 64.01 |

## 5 Related Work

**Class-imbalanced learning.** There are principally two overarching strategies to address class imbalance. Model-centric approaches try to handle the imbalance problem by modifying the objective function to alter the classifier margins (Cao et al., 2019; Tan et al., 2020; Menon et al., 2021), reweight minority classes (Japkowicz & Stephen, 2002; Cui et al., 2019), or correct the model in a posthoc fashion (Kang et al., 2020; Tian et al., 2020; Menon et al., 2021; Hong et al., 2021) by altering the logits in the inference phase. A data-centric approach is sampling, where synthetic minority samples are introduced into the training data via means such as augmentation (Kang et al., 2020; Liu et al., 2020; Ren et al., 2020) or generation (Chawla et al., 2002; Han et al., 2005; He et al., 2008; Kim et al., 2020; Chu et al., 2020; Zhang et al., 2021; Wang et al., 2021; Kim et al., 2022).

**Deep generative models for tabular data.** Although tabular data is a modality where deep learning does not generally excel over classical machine learning models, tabular generative models show distinctive advantages in modeling the generative distribution of the data. TVAE (Xu et al., 2019) is a variational autoencoder customized for tabular data. MedGAN (Choi et al., 2017), TableGAN (Park et al., 2018), CTGAN (Xu et al., 2019) are tabular generative model architectures based on generative adversarial networks. With the recent success of diffusion-based generative models (Ho et al., 2020; Song et al., 2021), the methodology has been successfully adapted to the tabular domain. TabDDPM (Kotelnikov et al., 2023) is based on denoising diffusion probabilistic models, while StaSy (Kim et al., 2023) is a score-based model. One unexpected tabular generative model is GReaT (Borisov et al., 2023), which is based on a generative language model.

## 6 Conclusion

In this paper, we presented a language-interfaced oversampling framework for tabular data that comprehensively utilizes the generation capabilities of generative language models. By progressively iterating class-conditioned borderline sampling and rejection sampling through self-authentication, our framework successfully generates synthetic minority samples that benefit the learning of machine learning classifiers. We verified the performance of our framework on multiple tabular datasets and imbalance ratios. Also, the proposed framework can be extended to black-box language models through in-context learning.

## ACKNOWLEDGEMENTS

This work was supported by Institute of Information & communications Technology Planning & Evaluation(IITP) grant funded by the Korea government(MSIT) (No.2019-0-00075, Artificial Intelligence Graduate School Program(KAIST), No.2022-0-00713, Meta-learning applicable to real-world problems, No.2022-0-00984, Development of Artificial Intelligence Technology for Personalized Plug-and-Play Explanation and Verification of Explanation).

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

## A    DETAILED EXPERIMENTAL SETTINGS

In this section, we describe the details of the experiments we conducted. We implement our method with the PyTorch deep learning framework and Huggingface Transformers package (Wolf et al., 2019). For the SMOTE and B-SMOTE baselines, we use the Imbalanced-Learn (Lemaître et al., 2017) package. For CTGAN and TVAE, we use the Synthetic Data Vault (Patki et al., 2016) package. For SOS and GReaT, we use their respective official Github repositories. For the machine learning models involved in measuring the Machine Learning Efficiency (MLE) metric, we use the Scikit-Learn (Pedregosa et al., 2011) package.

### A.1    DATASET DETAILS

In this subsection, we describe the 6 tabular benchmark datasets used in our experiments.

- `Default` : A binary classification dataset that provides information about default payment status among credit card clients in Taiwan.

- `Shoppers` : A binary classification task which involves predicting the intent of online shoppers.

- `Sick` : A binary classification task of medical domain that involves classifying thyroid disease as either negative or positive. For this, we exclude the rows which have the missing value.

- `Diabetes` : A binary classification dataset which contains information related to diabetes.

- `Obesity` : A multi-class classification task aiming to categorize obesity levels based on individuals' eating habits and physical condition.

- `SatImage` : A multi-class classification dataset, produced by the Australian Centre for Remote Sensing, designed for categorizing different types of land use.

Table 5: Details of dataset statistics used in our experiments.

| Datasets | #train | #test | #columns | #continuous | #categorical |
|----------|--------|-------|----------|-------------|--------------|
| Default  | 24K    | 6K    | 24       | 13          | 11           |
| Shoppers | 9.8K   | 2.4K  | 18       | 10          | 8            |
| Sick     | 1.7K   | 0.8K  | 29       | 7           | 22           |
| Diabetes | 0.6K   | 0.2K  | 10       | 7           | 3            |
| Obesity  | 1.6K   | 0.4K  | 25       | 24          | 1            |
| SatImage | 4.4K   | 1.3K  | 37       | 37          | 0            |

### A.2    HYPERPARAMETERS FOR EVALUATING MACHINE LEARNING EFFICIENCY

For reproducibility, we describe the hyperparameters we used for measuring machine learning efficiency.

- `Decision Tree` : Max depth is 32, criterion is gini.

- `AdaBoost` : The number of ensemble estimators is 100, and the learning rate is 1.0.

- `Logistic Regression` : We use "lbfgs" solver and 1000 max iteration. Penalty for training is $L2$ normalization.

- `MLP` : Used hidden layer size is (100, 100), max iteration is 200 and weight for penalty (alpha) is 0.0001.

- `Random Forest` : The number of estimators is 100 and the max depth is 8.

- `XGBoost` : We use the muti-label softmax objective, 5 max depth and 1.0 learning rate. The number of ensemble estimators is 100.

### A.3    Hyperparameters used for fine-tuning generative language models.

First, we utilize pre-trained generative language models from the established HuggingFace frame-work (Wolf et al., 2019). We fine-tune the Distill-GPT-2 model for each data set for 200 or 300 epochs according to convergence rate. If the loss curve is not converged, we fine-tune Distill-GPT-2 for 300 epochs, if not we fine-tune for 200 epochs. We use the constant 5e-5 learning rate following Borisov et al. (2023).

# B SAMPLING QUALITY AND DIVERSITY

Here, we analyze the sampling diversity and quality of LITO and other competing baselines. First, we conduct qualitative analysis by reporting the column-wise distributions of synthesized minority samples w.r.t. the ground truth minority samples. Also, we provide visualizations of the synthetic minority samples via UMAP (McInnes et al., 2018). For quantitative analysis, we report the coverage score (Naeem et al., 2020) to assess the quality and diversity of the synthesized samples.

## B.1 QUALITATIVE ANALYSIS

### B.1.1 COLUMN-WISE HISTOGRAMS

We first visualize the histogram of values for individual feature columns of synthetic minority samples generated by each method, compared to the real minority distribution. Figure 2, 3, 4 shows the histogram of column values on datasets used in our experiments. The results indicate that the samples generated by LITO are most close to the ground truth minority class distribution.

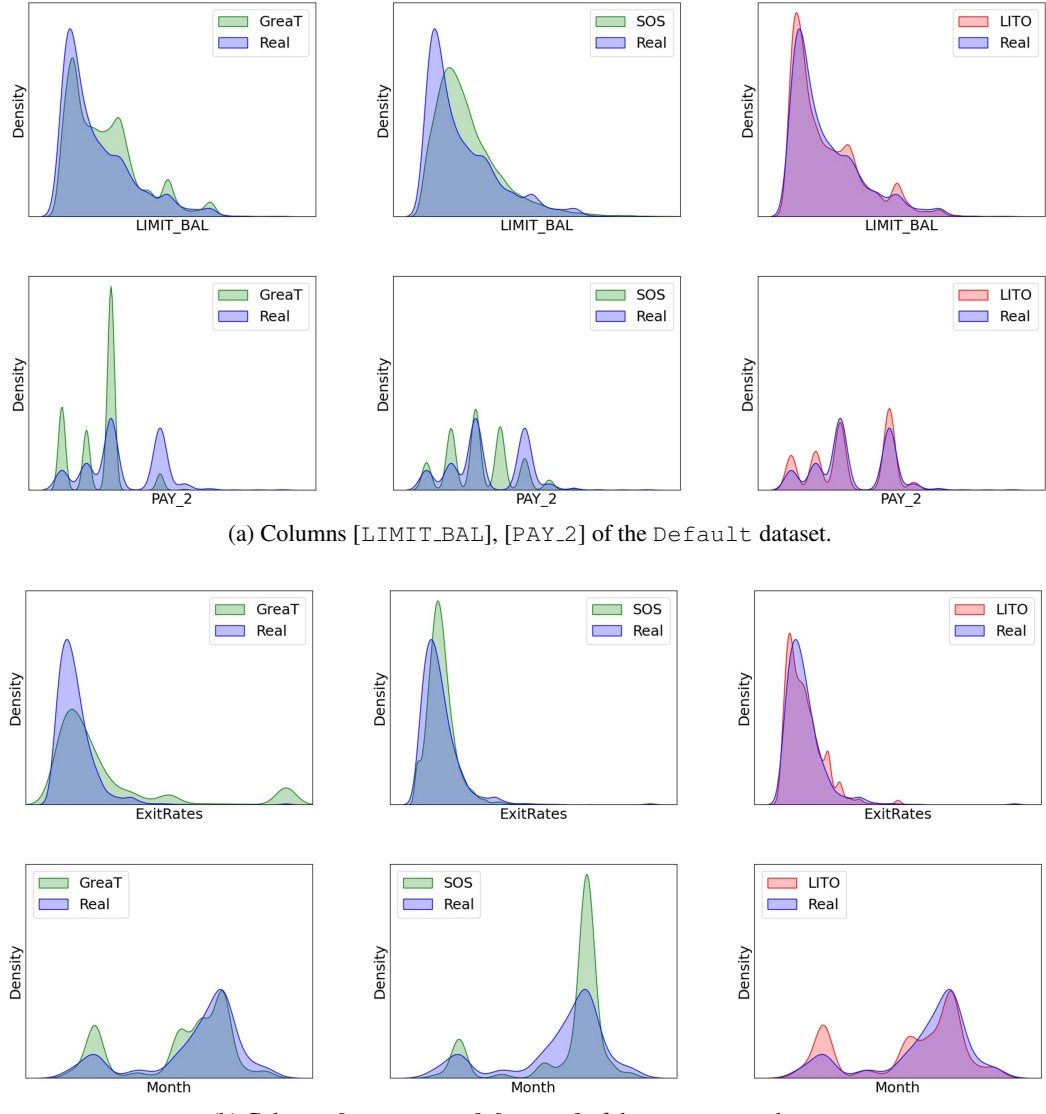

(a) Columns [LIMIT_BAL], [PAY_2] of the Default dataset.

(b) Columns [ExitRates], [Month] of the Shoppers dataset.

Figure 2: Histogram of column values of synthetic minority samples compared to the ground truth minority distribution.

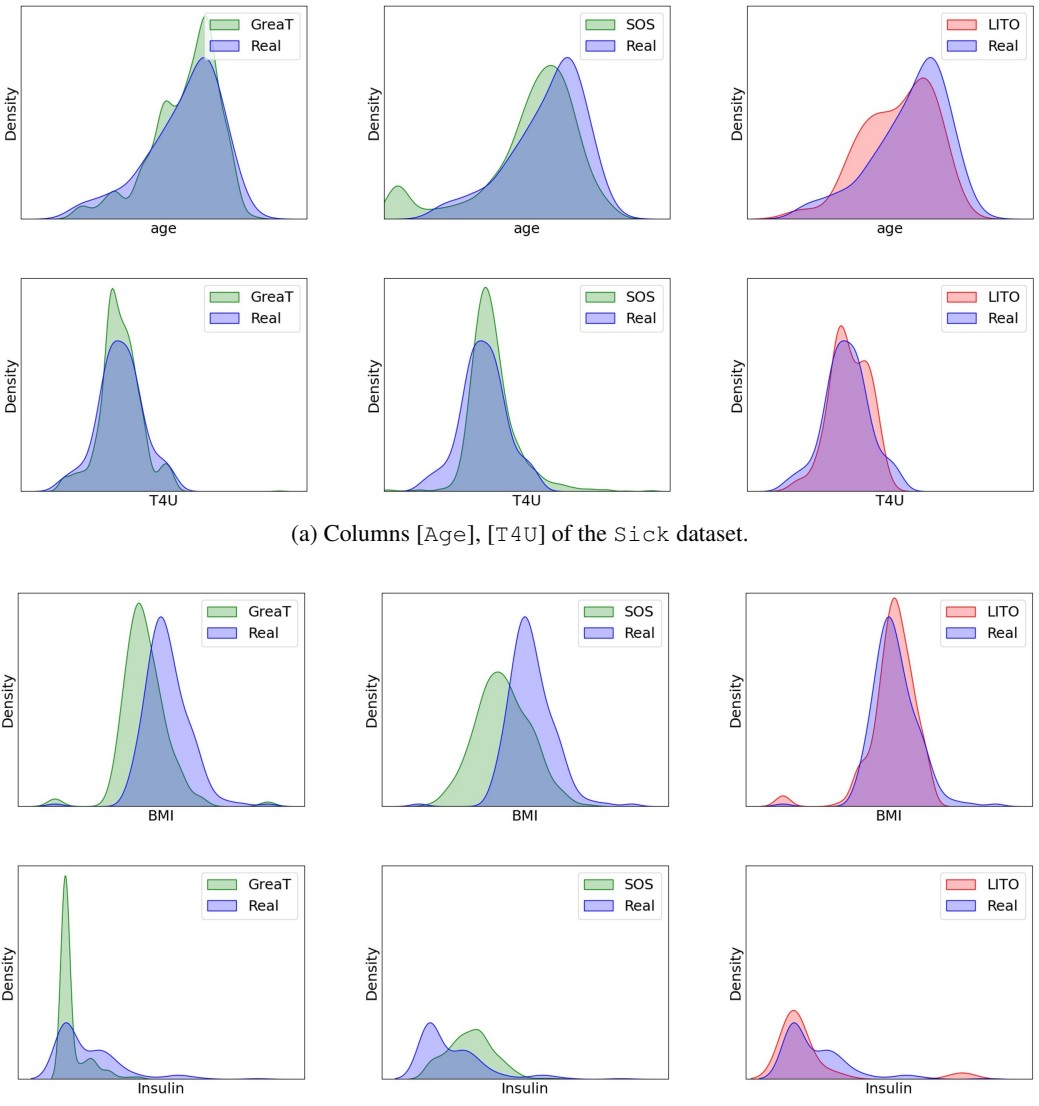

(a) Columns [Age], [T4U] of the Sick dataset.

(b) Columns [BMI], [Insulin] of the Diabetes dataset.

Figure 3: Histogram of column values of synthetic minority samples compared to the ground truth minority distribution.

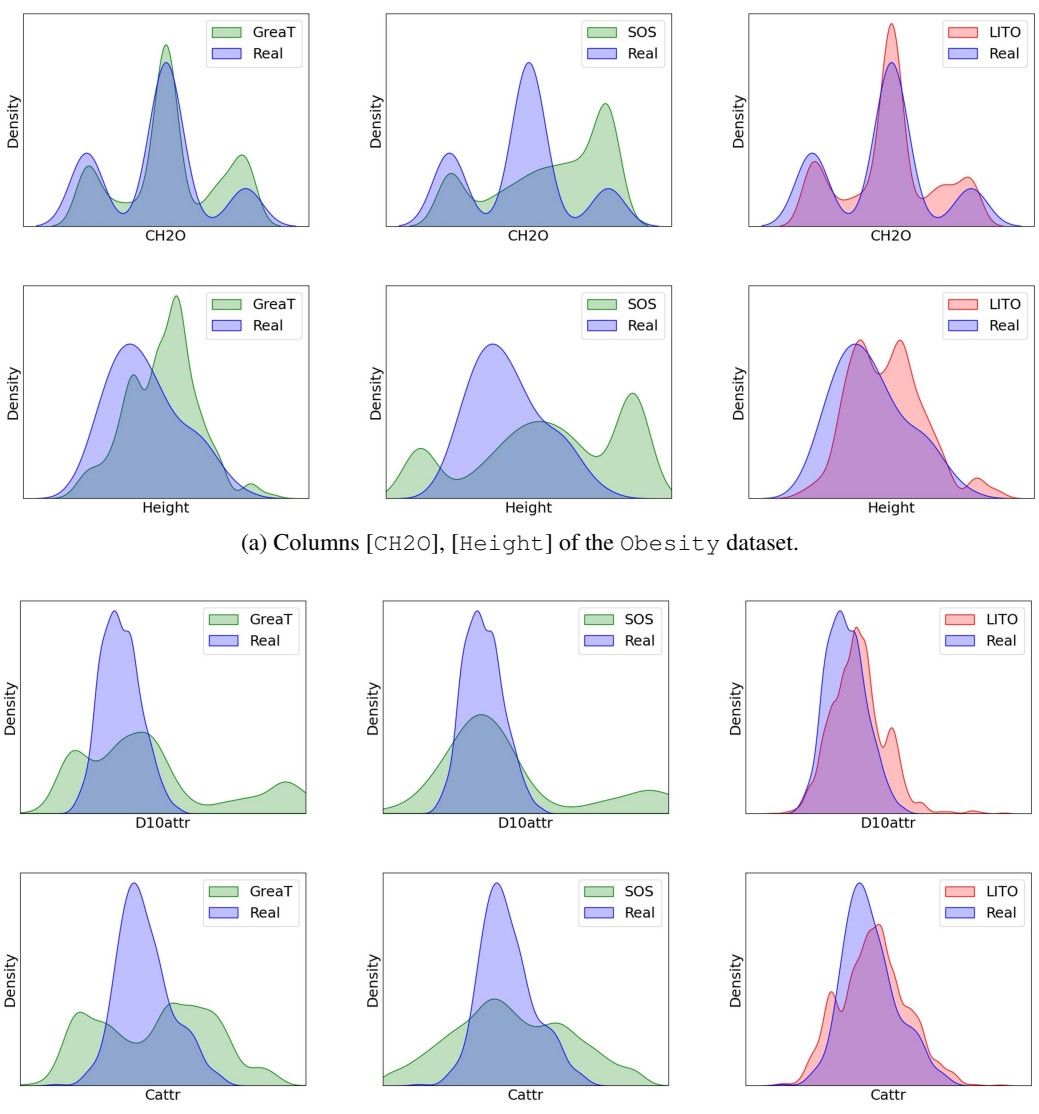

(a) Columns [CH2O], [Height] of the Obesity dataset.

(b) Columns [D10attr], [Cattr] of the Satimage dataset.

Figure 4: Histogram of column values of synthetic minority samples compared to the ground truth minority distribution.

### B.1.2 UMAP VISUALIZATIONS

Here, we visualize the manifold occupied by the synthetic minority samples via UMAP. Figure 5, 6 shows that the samples generated by LITO are most close to the ground truth minority class distribution.

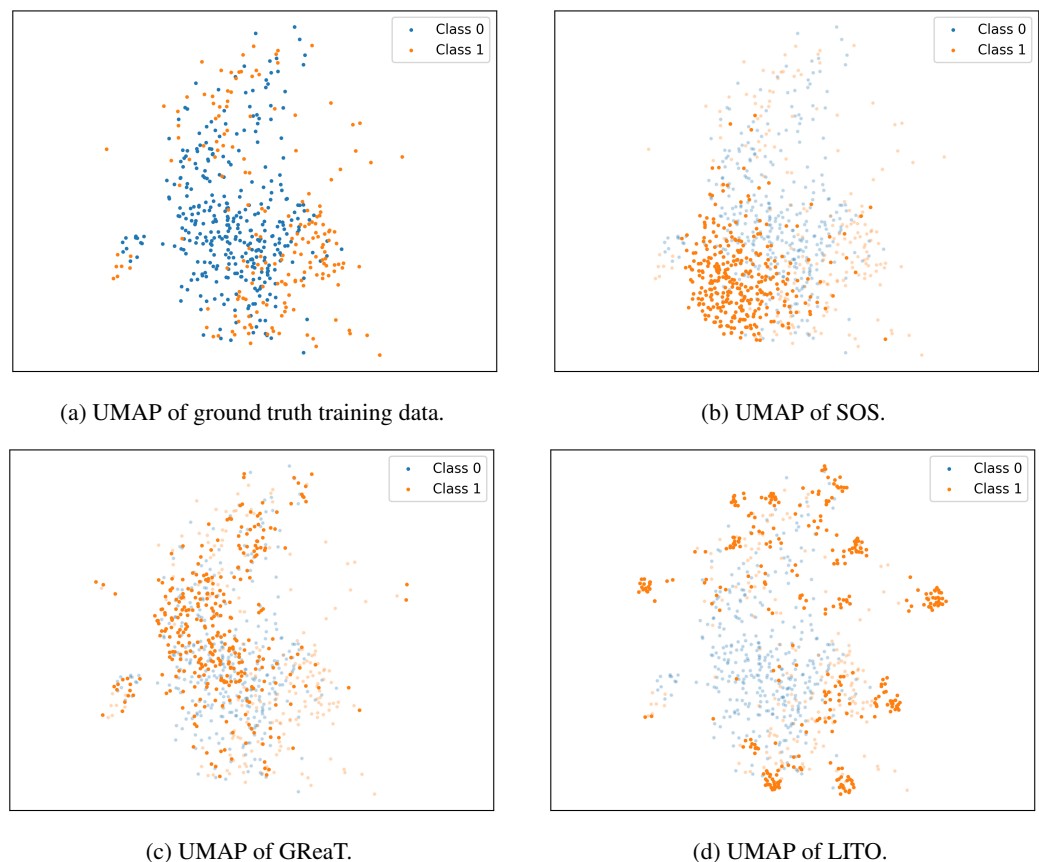

(a) UMAP of ground truth training data.

(b) UMAP of SOS.

(c) UMAP of GReaT.

(d) UMAP of LITO.

Figure 5: UMAP visualization results for for the `diabetes` dataset. For comparison, the ground truth training data is superimposed as transparent points. UMAP results show that while LITO successfully generates synthetic minority samples, while other baselines fall short in generating samples near the ground-truth minority distribution and overlap into the majority class distribution.

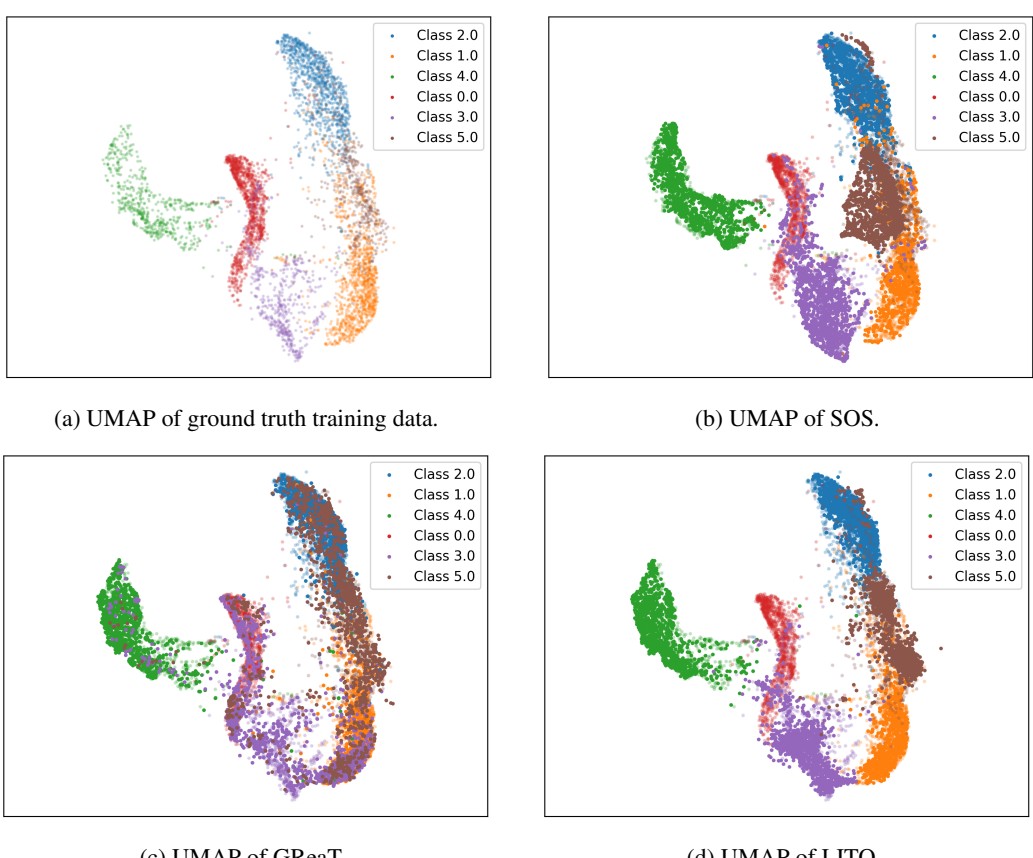

(a) UMAP of ground truth training data.

(b) UMAP of SOS.

(c) UMAP of GReaT.

(d) UMAP of LITO.

Figure 6: UMAP visualizations for the `satimage` dataset. For comparison, the ground truth training data is superimposed as transparent points. UMAP results show that LITO successfully generates synthetic minority samples that resemble the ground truth training data.

## B.2 QUANTITATIVE ANALYSIS

To quantitatively analyze the quality and diversity of generated samples, we measure the coverage score (Naeem et al., 2020), which measures whether at least one generated sample is contained in the $k$-nn manifold of a real data sample. Although a high coverage score reflects the diversity of the generated samples, it does not guarantee the quality of generated samples, as our main focus is in *minority oversampling*. For example, sampling distribution with a high coverage score may invade the majority class distribution, resulting in degraded downstream performance. In this sense, we devise and report the *spillage score* which is the coverage score of the sampled minority data w.r.t. the majority class samples. For minority oversampling, the generative model should obtain a high coverage score while maintaining a low spillage score. In Table 6 and Table 7, we report the respective scores of our method and recent competing baselines on binary and multi-class datasets for imbalance ratio 100.

Table 6: Coverage and spillage scores for binary classification tasks.

| | Imbalance ratio $\alpha = 100$ | | | | | | | |
| **Dataset** | Default | | Shoppers | | Sick | | Diabetes ($\alpha = 20$) | |
| | Coverage | Spillage | Coverage | Spillage | Coverage | Spillage | Coverage | Spillage |
|---|---|---|---|---|---|---|---|---|
| Real | 1 | 0.533 | 1 | 0.345 | 1 | 0.074 | 1 | 0.599 |
| SOS | 0.608 | 0.203 | 0.769 | 0.213 | 0.929 | 0.466 | 0.375 | 0.553 |
| GreaT | 0.859 | 0.883 | 0.969 | 0.884 | 0.943 | 0.457 | 0.798 | 0.931 |
| LITO-C | 0.945 | 0.629 | 0.891 | 0.324 | 0.950 | 0.466 | 0.761 | 0.338 |
| LITO-B | 0.951 | 0.656 | 0.870 | 0.440 | 0.936 | 0.485 | 0.676 | 0.376 |

Table 7: Coverage and spillage scores for multiclass classification tasks.

| | Imbalance ratio $\alpha = 100$ | | | |
| **Dataset** | Obesity | | Satimage | |
| | Coverage | Spillage | Coverage | Spillage |
|---|---|---|---|---|
| Real | 1 | 0.325 | 1 | 0.126 |
| SOS | 0.657 | 0.161 | 0.338 | 0.045 |
| GreaT | 0.847 | 0.392 | 0.630 | 0.400 |
| LITO-C | 0.912 | 0.370 | 0.672 | 0.108 |
| LITO-B | 0.926 | 0.386 | 0.509 | 0.125 |

## C    DETAILED SETTINGS FOR IN-CONTEXT LEARNING

For in-context learning oversampling experiments, we use the OpenAI chatbot API to access `GPT-3.5-turbo-0613`. For the generation step, we provide majority samples and minority samples as context and prompt the model to generate synthetic minority samples. Due to token limits, we supply 100 majority-class samples and 17 minority-class samples (all of them) in the context and request to generate samples or impute a given sample. The prompt structure used is described in Figure 7, 8. For LITO-B, For self-authentication, we use in-context few-shot (17 shots) learning to predict the labels of the generated samples. The prompt format used is described in Figure 9. In Table 8, we provide further experiments on ICL-LITO with and without self-authentication. Results show that self-authentication effectively increases the oversampling performance.

Table 8: In-context LITO results with and without self-authentication.

| Method | F1 | bAcc |
|---|---|---|
| Vanilla | 49.54 | 54.29 |
| SMOTE | 61.06 | 61.28 |
| B-SMOTE | 59.77 | 60.46 |
| CTGAN | 55.31 | 55.50 |
| SOS | 49.80 | 54.39 |
| GReaT (Distill-GPT2) | 49.44 | 49.59 |
| ICL-LITO-C (w/o SA) | 64.70 | 65.42 |
| ICL-LITO-C (with SA) | **67.37** | **66.88** |
| ICL-LITO-B (w/o SA) | 58.32 | 59.76 |
| ICL-LITO-B (with SA) | 63.68 | 64.01 |

| Role | Content |
|---|---|
| SYSTEM | You are a diabetes expert with a medical degree. |
| USER | This is a diabetes record csv file. These people are label 0.

Attr0,Attr1,Attr2,Attr3,Attr4,Attr5,Attr6,Attr7,label

2,106,56,27,165,29.0,0.426,22,0

1,117,60,23,106,33.8,0.466,27,0

…

5,128,80,0,0,34.6,0.144,45,0

This is an diabetes record csv file. These people are label 1.

Attr0,Attr1,Attr2,Attr3,Attr4,Attr5,Attr6,Attr7,label
0,135,68,42,250,42.3,0.365,24,1
1,122,64,32,156,35.1,0.692,30,1
…
0,129,110,46,130,67.1,0.319,26,1

Read given record csv files. Can you synthesize 40 label 1 records for me? |

Figure 7: In-context learning prompt format for LITO-C.

| Role | Content |
| --- | --- |
| SYSTEM | You are a diabetes expert with a medical degree. |
| USER | This is a diabetes record csv file. These people are label class0.

Pregnancies,Glucose,BloodPressure,SkinThickness,Insulin,BMI,DiabetesPedigreeFunction,Age,label

2,106,56,27,165,29.0,0.426,22,class0

1,117,60,23,106,33.8,0.466,27,class0

…

5,128,80,0,0,34.6,0.144,45,class0


This is an diabetes record csv file. These people are label class1.

Pregnancies,Glucose,BloodPressure,SkinThickness,Insulin,BMI,DiabetesPedigreeFunction,Age,label
0,135,68,42,250,42.3,0.365,24,class1
1,122,64,32,156,35.1,0.692,30,class1
…
0,129,110,46,130,67.1,0.319,26,class1


Now, let's look at the following class1 record. It has missing values, indicated by question marks:

Pregnancies,Glucose,BloodPressure,SkinThickness,Insulin,BMI,DiabetesPedigreeFunction,Age,label
0,?,?,0,0,?,0.262,?,class1

Generalizing from the previous records, impute the missing values and output directly into CSV format, starting with the column names. No talking. |

Figure 8: In-context learning prompt format for LITO-B.

| Role | Content |
| --- | --- |
| SYSTEM | You are a diabetes expert with a medical degree. |
| USER | Read a given information and questions. Think step by step, and then predict whether its value is class0 or class1. You must choose in [class0, class1].
Dataset has Pregnancies, Glucose, BloodPressure, SkinThickness, Insulin, BMI, DiabetesPedigreeFunction, Age as 8 input variables and label as output.
Question:When Pregnancies is 1, Glucose is 88, BloodPressure is 62, SkinThickness is 24, Insulin is 44, BMI is 29.9, DiabetesPedigreeFunction is 0.422, Age is 23, then what is the label? You must choose in [class0, class1]. Answer:class0
Question:When Pregnancies is 6, Glucose is 162, BloodPressure is 62, SkinThickness is 0, Insulin is 0, BMI is 24.3, DiabetesPedigreeFunction is 0.178, Age is 50, then what is the label? You must choose in [class0, class1]. Answer:class1

…

Question:When Pregnancies is 1, Glucose is 97, BloodPressure is 70, SkinThickness is 15, Insulin is 0, BMI is 18.2, DiabetesPedigreeFunction is 0.147, Age is 21, then what is the label? You must choose in [class0, class1]. Answer:class0
Question:When Pregnancies is 1, Glucose is 150, BloodPressure is 72, SkinThickness is 35, Insulin is 0, BMI is 33.5, DiabetesPedigreeFunction is 0.627, Age is 50, then what is the label? You must choose in [class0, class1]. Answer: |

Figure 9: In-context learning prompt format for self-authentication.

