# OpenReview forum: "Language-Interfaced Tabular Oversampling via Progressive Imputation and Self-Authentication"
_ICLR.cc/2024/Conference — ICLR 2024 poster_

### Official Review · Reviewer_tYE6 · 2023-10-30

**Soundness:** 2 fair
**Presentation:** 3 good
**Contribution:** 2 fair
**Rating:** 6
**Confidence:** 3

**Summary:**

This paper proposes a way to synthesize minority tabular samples using large language models in order to handle class-imbalance. A self-authentication framework is desiged that uses language models to predict the label of the generated sample and sifts out ill-converted samples. Experiments are conducted on a variety of datasets and imbalance ratios.

**Strengths:**

- Class-imbalance learning for tabular data is a under-exploited area. This paper makes a meaningful attempt.
- Synthesizing minor classess with large language models seems to be new.
- The proposed method is simple to implement. The conversion from tabular to textual representation and the self-authentification with LLM prediction can generalize to different LLMs.

**Weaknesses:**

- My biggest concern is on the reliability of those minor samples generated by LLMs. LLMs are shown to easily generate counter-factual texts. For the tabular data, although the authors use a self-authentification framework to filter out samples with incorrect prediction, there is no guarantee that the remaining samples are semantically meaningful. Considering the different feature columns, I doubt that current LLMs can truly understand the numeric feature values and their relative relation.
- The language interfacing requires column feature names. In many tabular data, feature names may not have meaningful semantics such as an abbreviation. It would be hard for LLMs to understand those features.

**Questions:**

- To understand LLM as a self-authenticator, I wonder the accuracy of directly using LLM to make classification on original data.

---

> ### Author Response · Authors · 2023-11-20
> **Response to Reviewer tYE6 (1)**
>
> We sincerely appreciate the reviewer tYE6’s constructive feedback. We address the reviewer’s concerns below.
>
> ---
>
> ### [W1, W2. Understanding the semantics of feature names.]
>
> As the reviewer has mentioned, the performance of LLMs may vary based on the semantic meaning of column feature names, as observed in [1]. However, even in the absence of column names, the language model does not break down in performance but rather is capable of processing the given data via its general sequential modeling capability originating from the pretrained weights when it is fine-tuned on the tabular dataset. For example, [2] leverages such capabilities to process time series data, which do not contain linguistic meaning.
>
> This capability is further observed in our main experiments. In the datasets used in our experiments, there are cases where column names do contain semantic meaning (e.g., Shoppers, Diabetes), contain semantics but lack context (e.g. Default), contains abbreviations (e.g., Sick, Obesity), or do not contain any meaning (e.g. Satimage). Below, we have listed selected column (feature) names for each dataset:
>
> Default: ['LIMIT_BAL', 'PAY_2', 'BILL_AMT1', 'PAY_AMT1', 'PAY_AMT2', …]
>
> Shoppers: ['ExitRates', 'PageValues', 'SpecialDay', 'Month', 'VisitorType', …]
>
> Diabetes: [’Glucose’, ‘BloodPressure’, ‘Insulin’, ‘BMI’, ‘Age’, …]
>
> Sick: [’TSH’, ‘T3’, ‘TT4’, ‘T4U’, ‘FTI’, …]
>
> Obesity: [’FAVC’, ‘NCP’, ‘CAEK’, ‘CH2O’, ‘FAF’, … ]
>
> Satimage: [Aattr, Cattr, A1attr, D10attr, D28attr, …]
>
> Even presented with a wide and versatile range of semantics levels for feature names, the main experiments in Section 4 show that our method surpasses other methods in the oversampling task. In Appendix C.1.1 of the revised manuscript, we provide the feature-wise histogram visualizations of synthetic samples w.r.t. the real distribution, where our method exceeds other baselines in modeling individual columns of varying levels of semantics (e.g. PAY_2, Month, T4U, D10attr, Cattr), even for datasets such as Satimage where feature names are devoid of meaning.
>
> ---
>
> ### [W1. Reliability of the synthesized samples.]
>
> Since the backbone tabular language model(TLM) of LITO is first fine-tuned on a given tabular dataset, the TLM can successfully learn the given tabular data and the relative relations of columns, as shown in [1, 3]. However, as the reviewer has mentioned, recent observations in language models have shown that they can generate counterfactual texts. This phenomenon is exacerbated when the availability of minority-class samples is low, which is verified in our experimental observation where a TLM baseline(GReaT) produces samples that intrude into the distribution of other classes even when it is prompted to generate a certain minority class, which can be considered as counterfactual generation. This is in fact why we employ self-authentication to filter out the counterfactual samples. Moreover, our importance-aware imputation method only re-samples the important columns, keeping other columns in the real data distribution - reducing counterfactual noise.
>
> Visualizations in Appendix C.1.2 show that our method effectively reduces counterfactual generation(intrusion into other classes) compared to baselines.  Quantitative analyses in Appendix C.2 indicate that LITO produces meaningful samples(coverage) while suppressing counterfactual samples(spillage).
>
> ---
>
> ### [Q1. classification accuracy of the language model on original data.]
>
> In the table below, we report the classification performance of the backbone TLM on the test dataset:
>
> | | F1 | bACC |
> | --- | --- | --- |
> | Default | 50.41 | 52.82 |
> | Shoppers | 60.63 | 62.35 |
> | Sick | 67.51 | 68.20 |
> | Diabetes | 54.24 | 53.02 |
> | Obesity | 51.58 | 52.14 |
> | Satimage | 75.15 | 76.21 |
>
> ---
>
> [1] Dinh et al. "LIFT: Language-interfaced fine-tuning for non-language machine learning tasks." NeurIPS 2022.
>
> [2] zhou et al. “One Fits All: Power General Time Series Analysis by Pretrained LM.” NeurIPS 2023.
>
> [3] Borisov et al. "Language models are realistic tabular data generators." ICLR 2023.

---

> > ### Comment · Reviewer_tYE6 · 2023-11-22
> > **Reply to author rebuttal**
> >
> > I appreciate the authors' response on my previous concerns. I am still not quite convinced by the explaination "...but rather is capable of processing the given data via its general sequential modeling capability" and the example with time series data. To me, time series data have an inherent causality between previous and current timesteps, while columns of tabular data are disordered.
> >
> > Given the provided evidence that the propose method can exceed other baselines and reduce counterfactual generation, I decided to raise my score to be positive.

---

> > > ### Author Response · Authors · 2023-11-23
> > >
> > > We first thank the reviewer for reviewing our response and raising the score of our paper. We sincerely appreciate your time and consideration. We further address the reviewer’s concerns below.
> > >
> > > ---
> > >
> > > The sequential modeling capability of language models results in the capability of modeling a column given previous columns, which is utilized in generating tabular data. However, as the reviewer has mentioned, the columns of tabular data are disordered. To this end, column order random permutation augmentation[3] is used in fine-tuning to promote column order independence.

---

### Official Review · Reviewer_gyRK · 2023-11-02

**Soundness:** 3 good
**Presentation:** 2 fair
**Contribution:** 2 fair
**Rating:** 6
**Confidence:** 3

**Summary:**

This paper studies the class imbalance problem in tabular data processing. Specifically, one major drawback of class imbalance in tabular data is that the minority data is not sufficient to help learn a good predictor. In order to solve this problem, the authors propose to leverage the language model to synthesize minority data. Specifically, there are three major parts, first, the minorities are generated by conditioning on their class labels; then, some incorrect examples are rejected through self-authentication; finally, the authors propose an adaptive oversampling by progressively imputation. Through extensive quantitative experiments, the proposed method shows a larger performance improvement than other baseline methods.

**Strengths:**

- This paper is well-written and well-organized.
- Experimental validations are extensive and effective over other methods.
- The minority generation can effectively solve the class imbalance problem.

**Weaknesses:**

- There have already been many methods that leverage language models to augment data. Using a similar strategy to solve the class imbalance problem seems to be a trivial contribution to me. Could the authors identify why the proposed method could be a novel contribution?
- Whether the generation of minority data could help the learning performance is still unclear to me. It would be better if the quality of the synthesized data could be carefully evaluated. Moreover, how many examples are rejected during self-authentication? It would further improve the quality of this paper if a proper ablation study were conducted.
- How can different language models influence the performance of the proposed method? Could the author try it on GPT-3 or ChatGPT?
- Minor: The Algorithm is taking up too many blank spaces.

**Questions:**

Please refer to the weaknesses.

**Details Of Ethics Concerns:**

No ethics concers

---

> ### Author Response · Authors · 2023-11-20
> **Response to Reviewer gyRK (1)**
>
> We sincerely thank the reviewer gyRK for their insightful comments. We address the reviewer's concerns below.
>
> ---
>
> ### [W1. Novel contribution of the proposed method.]
>
> While data augmentation strategies via language models are on the rise, our work focuses on whether language models are practically applicable for tabular data generation under class-imbalanced situations, which are frequent in the tabular domain. To this end, we propose a novel oversampling method with two components: progressive importance-aware imputation and self-authentication. In comparison to prior works, our paper introduces novel contributions in both its findings and methodology, which include:
>
> 1. Experimental observations (Section 4.2, 4.3, and Appendix C of the revised manuscript) show that naive oversampling with a tabular language model(GReaT) fine-tuned on imbalanced data to oversample minority class samples results in low-quality samples that degrade downstream classification performance(as in samples that overlap into the distributions of other classes), due to the scarcity of minority class samples.
> 2. To filter these samples and enhance sample quality, we propose self-authentication, which utilizes the language model itself. Our experiments in Section 4.4 show that self-authentication is an effective component for increasing the quality of the generated samples, which is reflected in the increase in downstream classification performance.
> 3. We propose progressive importance-aware imputation, a novel borderline sampling strategy that integrates the domain characteristics of tabular data and the capabilities of language models.
>     * Unlike other data modalities(such as vision, speech, …) where class-relevant information is spread across features, the features of tabular data each hold distinctive information, where there exists columns important to the given task, and also relatively unimportant columns[1]. However, previous borderline sampling strategies such as [2, 3] manipulate all features simultaneously - introducing potential noise in the unimportant columns thereby degrading sample quality.
>     * Using this domain characteristic, we propose to iteratively select, puncture, and re-impute the important columns, using the conditional sampling capability of the language model. This ensures the feature quality of the unimportant columns as they stay in the real distribution, while important columns are resampled for diversity.
>     * To identify the instance-wise important columns,  we propose to leverage the self-attention maps of the language model to measure the feature importance.

---

> ### Author Response · Authors · 2023-11-20
> **Response to Reviewer gyRK (2)**
>
> ### [W2. Quality of the synthesized data.]
>
> As observed in our main experiments(Section 4.2, 4.3) of the main paper, the classification performance increase of various downstream machine learning classifiers (logistic regression, xgboost, ... etc.) augmented with LITO oversampling reflects that our method generates quality samples that are effective in increasing classification performance compared to other baselines.
>
> To further examine the qualities of the synthesized data, we undertake both qualitative and quantitative analysis of synthesized samples. The detailed figures and descriptions are provided in Appendix C.1 and C.2 of the revised manuscript.
>
> In our qualitative analysis, following the methodology of previous work[2], we present **feature-wise histograms** and **UMAP**[4] **visualization** plots for synthesized samples for various datasets, which are included in Appendix C.1. These histograms reveal that LITO's column features more closely match the actual minority distribution than other comparison baselines, demonstrating the quality of the samples synthesized with our method. Additionally, we construct UMAP diagrams to visually compare the manifolds occupied by both the training data and the synthetic minorities created by our method and other baselines. The results demonstrate the ability of LITO to accurately represent the true minority distribution while avoiding intrusion into the distribution of other classes.
>
> To quantitatively assess the quality of synthesized samples, we employ the Coverage Score[5] and measure the overlap of synthetic minority samples with the real minority samples. However, focusing solely on coverage w.r.t. the real minority samples may neglect the fact that naive high coverage can lead to overlap with the majority class distributions, reducing the effectiveness of minority oversampling and adversely affecting the performance of downstream classification tasks. Consequently, we also calculated the coverage score between the synthesized minorities and real majority samples, a metric we refer to as *"spillage"* temporarily. For relative reference, we also measure the natural spillage of real minority data into the real majority data and report its difference. The provided table contrasts the coverage and spillage scores of LITO with those of recent baselines. LITO demonstrates a balance of high coverage and low spillage, whereas GReaT exhibits significant spillage and SOS has limited coverage. These findings suggest that our approach not only generates a diverse set of samples but also maintains their quality. This pattern is also observed in the UMAP visualizations in Appendix C.1.
>
> | imb = 100 | Default |  |  | Shoppers |  |  | Sick |  |  | Diabetes |  |  |
> | --- | :---: | :---: | :---: | :---: | :---: | :---: | :---: | :---: | :---: | :---: | :---: | :---: |
> |  | Coverage(↑) | Spillage(↓) | ΔSpill | Coverage | Spillage | ΔSpill | Coverage | Spillage | ΔSpill | Coverage | Spillage | ΔSpill |
> | Real | 1 | 0.533 | - | 1 | 0.345 | - | 1 | 0.074 | - | 1 | 0.599 | - |
> | SOS | 0.608 | 0.203 | 0.330 | 0.769 | 0.213 | 0.131 | 0.929 | 0.466 | 0.392 | 0.375 | 0.553 | 0.046 |
> | GReaT | 0.859 | 0.883 | 0.350 | 0.969 | 0.884 | 0.539 | 0.943 | 0.457 | 0.383 | 0.798 | 0.931 | 0.332 |
> | LITO | 0.951 | 0.656 | 0.123 | 0.870 | 0.440 | 0.095 | 0.936 | 0.485 | 0.411 | 0.676 | 0.376 | 0.226 |
>
> | imb = 100 | Obesity |  |  | Satimage | (*LITO-C) |  |
> | --- | :---: | :---: | :---: | :---: | :---: | :---: |
> |  | Coverage | Spillage | ΔSpill | Coverage | Spillage | ΔSpill |
> | Real | 1 | 0.325 | - | 1 | 0.126 | - |
> | SOS | 0.657 | 0.161 | 0.164 | 0.338 | 0.045 | 0.082 |
> | GReaT | 0.847 | 0.392 | 0.067 | 0.630 | 0.400 | 0.273 |
> | LITO | 0.926 | 0.386 | 0.061 | 0.672 | 0.108 | 0.019 |
>
> To further address the reviewer’s concerns, we record the acceptance ratios of self-authentication. The table below shows the sample acceptance ratio for the datasets used in the main experiment. As the imbalance ratio increases, the acceptance ratio is lower, maintaining the sample quality.
>
> | Imb. ratio | Diabetes | Default | Shoppers | Sick  | Obesity | Satimage |
> | --- | :---: | :---: | :---: | :---: | :---: | :---: |
> | 10 | 55.48 | 20.28 | 51.30 | 93.31 | 67.24 | 41.98 |
> | 100 | 1.04 | 13.83 | 12.95 | 14.72  | 10.61 | 8.28 |

---

> ### Author Response · Authors · 2023-11-20
> **Response to Reviewer gyRK (3)**
>
> ### [W3. Using different language models e.g. GPT-3 or ChatGPT.]
>
> Since the general capabilities of language models increase with their number of parameters, their capability to model tabular data can also increase with model size, as observed in [6, 7]. Since our method utilizes a language model backbone, the performance of our method can be further improved when equipped with a larger language model. This is observed in Section 4.5 of the main paper, where we adopt LITO to ChatGPT-3.5-turbo via in-context learning (ICL) and conduct experiments on the Diabetes dataset for imbalance ratio 20. We briefly summarize the results in the table below:
>
> | Method | F1 | bAcc |
> | --- | --- | --- |
> | Vanilla | 49.54 | 54.29 |
> | SMOTE | 61.06 | 61.28 |
> | BSMOTE | 59.77 | 60.46 |
> | CTGAN | 55.31 | 55.50 |
> | SOS | 49.80 | 54.39 |
> | GReaT(DistillGPT2) | 49.44 | 49.59 |
> | Ours(DistillGPT2) | 63.04 | 63.20 |
> | Ours(ChatGPT) | **67.37** | **66.88** |
>
> Experimental results show that utilizing larger language models results in a further performance increase of our method.
>
> ---
>
> [1] Grinsztajn et al. "Why do tree-based models still outperform deep learning on typical tabular data?." NeurIPS 2022.
>
> [2] Kim et al. "SOS: score-based oversampling for tabular data." KDD 2022.
>
> [3] Han et al. “Borderline-smote: a new over-sampling method in imbalanced data sets learning.” ICIC 2005.
>
> [4] McInnes et al. UMAP: Uniform Manifold Approximation and Projection. Journal of Open Source Software, 2018.
>
> [5] Naeem et al. "Reliable fidelity and diversity metrics for generative models." ICML 2020.
>
> [6] Dinh et al. "LIFT: Language-interfaced fine-tuning for non-language machine learning tasks." NeurIPS 2022.
>
> [7] Borisov et al. "Language models are realistic tabular data generators." ICLR 2023.

---

> > ### Comment · Reviewer_gyRK · 2023-11-22
> > **Reply to author rebuttal**
> >
> > Thanks for the thorough reply to all my questions. The explanations are reasonable and additional experiments show promising results. Hence, I decided to raise my score to 6.

---

> ### Author Response · Authors · 2023-11-23
>
> Thank you for your valuable insights and reconsideration of scores. We are pleased that the explanations and additional experiments have addressed the reviewer’s concerns. We sincerely appreciate your time in reviewing our response and paper.

---

### Official Review · Reviewer_puAc · 2023-11-05

**Soundness:** 3 good
**Presentation:** 4 excellent
**Contribution:** 3 good
**Rating:** 6
**Confidence:** 3

**Summary:**

This paper targets at the problem that tabular data usually struggles with class-imbalance problem.
To deal with this issue, this paper introduces an algorithm which oversamples the tabular data in a self-authentication manner.
Specifically, the algorithm synthesizes the minority class samples by progressively masking the important features of the majority class samples and imputing them towards the minority distribution.
The experiments show that the algorithm is able to generate reliable minority which can boost the ML classifier performance.

**Strengths:**

- The writing is good enough to make readers easily understand the paper. For example, Figure 1 and algorithm covers most of the details in the algorithm, which can make readers briefly know the whole algorithm in a short period."
- The targeted problem, minority in the tabular data, is interesting and still an under-presented problem in the community. Improving this problem can make a large improvement on the ML community.

**Weaknesses:**

- The main concern is that the oversampling algorithm is based on masking. While some of the information is changed during masking and self-authentication, most of the information still remain and suggests the algorithm just repeat some similar data and limit the diversity of data. Have you tried to analysis the label distribution between real negatives, synthesized negatives, and real positives?
- Another concern s about that the novelty of the proposed method. While the claimed contribution in this paper include "data generation" with "self-authentication", the similar ideas have appeared in some literature, like [1] and [2]. Could the authors state the difference between this work and the previous?

[1] Language Models are Realistic Tabular Data Generators, ICLR 2023

**Questions:**

- As mentioned in the weakness section, have you tried to analysis the label distribution between real negatives, synthesized negatives, and real positives? Can this algorithm bring the dataset diversity which can bring the benefit to the downstream task?

---

> ### Author Response · Authors · 2023-11-20
> **Response to Reviewer puAc (1)**
>
> We sincerely thank reviewer puAc for constructive feedbacks. We address the reviewer's concerns below.
>
> ---
>
> ### [W1, Q1. Diversity of the generated synthetic minority samples.]
>
> In order to examine the diversity of the synthetic minority samples generated by LITO(ours), we conducted additional qualitative and quantitative analyses, where detailed figures and descriptions are added in Appendix C of the revised manuscript.
> For qualitative analyses, we follow previous work[1] and plot the **individual feature-wise histograms** and **UMAP**[2] **visualizations** of synthesized minority samples for multiple datasets. The resulting figures are presented in Appendix C.1.1 and C.1.2, respectively. The individual feature-wise histograms show that the column features generated by LITO best resemble the ground truth minority distribution compared to other baselines, showcasing sampling diversity and quality. Also, we plot UMAP diagrams to visually observe the manifolds occupied by the training data and synthetic minorities generated by our method and other baselines. As seen in the figures, our method effectively covers the ground truth minority distribution while refraining from intruding into the distribution covered by other classes.
>
> To quantitatively measure the diversity of the synthesized minority samples compared to the ground truth minority distribution, we measure the Coverage Score[3] between the real minority samples and synthesized minority samples (Appendix C.2). However, measuring coverage **only** against the real minority samples neglects the quality aspect of the synthetic samples, as a distribution with high coverage only may simply ‘spill’ into the distribution of majority class samples, degrading minority oversampling quality thereby lowering downstream classification performance. To this end, we also measure the Coverage score between **real majority samples** and synthesized minorities (for convenience, we temporarily term this as *"spillage"*). For relative reference, we also measure the natural spillage of real minority data into the real majority data and report its difference. The table below shows the coverage and spillage scores of LITO and recent competing baselines. LITO exhibits high coverage while maintaining low spillage, while GReaT shows high spillage, and SOS shows low coverage. These results indicate that our method generates diverse samples while maintaining quality. This tendency is also visually confirmed in the UMAP visualizations in Appendix C.1.2.
>
> | imb = 100 | Default |  |  | Shoppers |  |  | Sick |  |  | Diabetes |  |  |
> | --- | :---: | :---: | :---: | :---: | :---: | :---: | :---: | :---: | :---: | :---: | :---: | :---: |
> |  | Coverage(↑) | Spillage(↓) | ΔSpill | Coverage | Spillage | ΔSpill | Coverage | Spillage | ΔSpill | Coverage | Spillage | ΔSpill |
> | Real | 1 | 0.533 | - | 1 | 0.345 | - | 1 | 0.074 | - | 1 | 0.599 | - |
> | SOS | 0.608 | 0.203 | 0.330 | 0.769 | 0.213 | 0.131 | 0.929 | 0.466 | 0.392 | 0.375 | 0.553 | 0.046 |
> | GReaT | 0.859 | 0.883 | 0.350 | 0.969 | 0.884 | 0.539 | 0.943 | 0.457 | 0.383 | 0.798 | 0.931 | 0.332 |
> | LITO | 0.951 | 0.656 | 0.123 | 0.870 | 0.440 | 0.095 | 0.936 | 0.485 | 0.411 | 0.676 | 0.376 | 0.226 |
>
> | imb = 100 | Obesity |  |  | Satimage | (*LITO-C) |  |
> | :---: | :---: | :---: | :---: | :---: | :---: | :---: |
> |  | Coverage | Spillage | ΔSpill | Coverage | Spillage | ΔSpill |
> | Real | 1 | 0.325 | - | 1 | 0.126 | - |
> | SOS | 0.657 | 0.161 | 0.164 | 0.338 | 0.045 | 0.082 |
> | GReaT | 0.847 | 0.392 | 0.067 | 0.630 | 0.400 | 0.273 |
> | LITO | 0.926 | 0.386 | 0.061 | 0.672 | 0.108 | 0.019 |
>
> We also analyze the label distribution of synthesized samples. Since we cannot know the oracle true labels for synthetic samples, we approximately infer their labels using an XGBoost classifier trained using the full training data. Since the test accuracy of the XGBoost classifier is not 100%, there exist potential discrepancies. The table below shows the inferred label distribution: the percentage of synthesized samples produced with our method and competing baselines that are predicted as the minority class, where our method surpasses or is on par with other baselines.
>
> XGBoost bAcc: 71.19(Default), 78.83(Shoppers), 82.61(Sick), 68.23(Diabetes)
>
> | imb = 100 | Default | Shoppers | Sick | Diabetes |
> | --- | :---: | :---: | :---: | :---: |
> | SOS | 42.38 | 62.59 | 66.01 | 41.24 |
> | GReaT | 40.78 | 56.35 | 68.40 | 39.94 |
> | Ours | 46.51 | 61.77 | 70.84 | 50.5 |
>
> Finally, as observed in our main experiments(Section 4.2, 4.3) of the main paper, the increase in classification performance of various downstream machine learning classifiers (logistic regression, xgboost, ... etc.) trained with LITO-oversampled data also reflects the quality and diversity of the synthetic minority samples generated by our method, compared to other baselines.

---

> ### Author Response · Authors · 2023-11-20
> **Response to Reviewer puAc (2)**
>
> ### [W2. Novelty of the proposed method.]
>
> Although [4] has demonstrated the possibility of language models for the generation of tabular data, our work focuses on the practical applicability of such tabular language models - whether they can be effectively adopted for oversampling to remedy the class-imbalance problem, which is a frequent and substantial issue in the tabular domain. To this end, we propose a novel sampling strategy comprised of two components: progressive importance-aware imputation and self-authentication. Compared to previous works, our proposed work contains the following novel contributions:
>
> 1. Our experimental observations, including Section 4.2, 4.3, and Appendix C of the revised paper, show that naive oversampling with a tabular language model(GReaT) fine-tuned on imbalanced data to oversample minority class samples results in ill-generated samples (for example, synthetic samples that intrude into distributions of other classes) that degrade downstream classification performance, due to the scarcity of minority class samples.
> 2. To filter out such samples, we propose self-authentication, which utilizes the capabilities of the language model itself. Our experiments in Section 4.4 show that self-authentication serves as an effective filter for maintaining the quality of the tabular samples generated by a tabular language model.
> 3. We propose progressive importance-aware imputation, a novel borderline sampling strategy that combines and leverages the characteristics of the tabular domain and the capabilities of language models simultaneously.
>     * Unlike other data modalities(e.g. vision, speech) where class-relevant information is spread across features, the column features of tabular data each hold distinctive information, where there exist columns important to the given task, and also relatively unimportant columns[5]. However, previous borderline sampling strategies such as [1, 6] manipulate all features simultaneously - introducing potential noise in the unimportant columns thereby degrading sample quality.
>     * Inspired by this characteristic, we propose to select, puncture, and re-impute the important columns iteratively, using the conditional sampling capability of the language model. This ensures the feature quality of the unimportant columns as they stay in the real distribution, while important columns are resampled for diversity.
>     * We propose to utilize the inherent self-attention maps of the language model to identify and measure the importance of column features instance-wise.
>
> ---
>
> Due to possible technical issues with the website, citation [2] mentioned by the reviewer in the review text is not visible to us. To this end, we politely request the reviewer to re-inform us of the name of the paper for citation [2] in the review.
>
> ---
>
> [1] Kim et al. "SOS: score-based oversampling for tabular data." KDD 2022.
>
> [2] McInnes et al. UMAP: Uniform Manifold Approximation and Projection. Journal of Open Source Software, 2018.
>
> [3] Naeem et al. "Reliable fidelity and diversity metrics for generative models." ICML 2020.
>
> [4] Borisov et al. "Language models are realistic tabular data generators." ICLR 2023.
>
> [5] Grinsztajn et al. "Why do tree-based models still outperform deep learning on typical tabular data?." NeurIPS 2022.
>
> [6] Han et al. “Borderline-smote: a new over-sampling method in imbalanced data sets learning.” ICIC 2005.

---

> > ### Comment · Reviewer_puAc · 2023-11-23
> >
> > Thanks authors for the response. The new results show the diversity of the generated synthetic minority samples. I will maintain my current rating for acceptance.

---

> ### Author Response · Authors · 2023-11-23
>
> Thank you for reviewing our response. We sincerely appreciate your time and effort in reviewing our paper and giving constructive feedback.

---

### Author Response · Authors · 2023-11-20
**General Response**

We thank the reviewers for their constructive feedback. Based on their comments, we revised our paper by making the following changes, which are colored in blue:

Major updates
* We have included analyses on the quality and diversity of generated samples in Appendix C.

Minor updates
* Added minor elaborations in the algorithm table and the introduction section.
* We also corrected minor typos and errors.

We thank the reviewers again for their insights.

Sincerely,

Authors

---

### Meta-Review · Area_Chair_BsUx · 2023-12-06

**Metareview:**

This is a timely paper that uses LLMs for oversampling minority classes in tabular data.
In contrast to existing work, the method introduced here also uses a "self-authentication" process, in which the LLM judges samples and filters out poor ones.
All three reviewers engaged with the authors and two of them also raised their score, leading to scores of 6,6,6, all borderline accepts.
It is unfortunate that the paper does not make an implementation available (at least it does not include the code or link to it). Inavailability of code would substantially limit its impact. Overall, the paper is borderline. I strongly encourage the authors to include code if accepted.

**Justification For Why Not Higher Score:**

The novelty is not very high, due to the existence of related methods, like "Language Models are Realistic Tabular Data Generators". (Yes, the paper still has *some* novelty, as the authors mention.)

**Justification For Why Not Lower Score:**

Handling imbalanced data in tabular classification is important, and this paper proposes a general method for doing so, with strong performance.

---

### Decision · Program_Chairs · 2024-01-16

Accept (poster)